# Don't stop me now: Psychological effects of interrupting a moving pedestrian crowd and a video game

**Ezel Üsten**[1,2]*, **Anna Sieben**[3]

**1** Civil Safety Research (IAS-7), Jülich Research Center, Jülich, North Rhine-Westphalia, Germany,
**2** Department of Computer Simulation for Fire Safety and Pedestrian Traffic, Wuppertal University, Wuppertal, North Rhine-Westphalia, Germany, **3** Department of Cultural and Social Psychology, University of St. Gallen, St. Gallen, Switzerland

* e.uesten@fz-juelich.de

**Citation:** Üsten E, Sieben A (2023) Don't stop me now: Psychological effects of interrupting a moving pedestrian crowd and a video game. PLoS ONE 18(7): e0287583. https://doi.org/10.1371/journal.pone.0287583

**Data Availability Statement:** The data from the studies presented in the manuscript are fully accessible in the Forschungszentrum Jülich - Pedestrian Dynamics Data Archive through the following links and respective DOIs: Study 1: https://ped.fz-juelich.de/da/doku.php?id=

## Abstract

Interruptions are a part of our everyday lives. They are inevitable in complex societies, especially when many people move from one place to another as a part of their daily routines. The main aim of this research is to understand the effects of interruptions on individuals from a psychological and crowd dynamics perspective. Two studies were conducted to investigate this issue, with each focusing on different types of interruptions and examining their psychological (emotion, motivation, arousal) and physiological (heart rate) components. Study 1 examined interruptions in a video game setting and systematically varied goal proximity (N = 61). It was hypothesized that being interrupted in the later stages of goal pursuit would create a high aroused impatience state, while interruptions in the earlier stages would produce a low aroused boredom state. However, the results showed that the hypothesized groupwise differences were not observed. Instead, interruptions created annoyance in all conditions, both psychologically and physiologically. Study 2 investigated interruptions in pedestrian crowds (N = 301) and used a basic motivational dichotomy of high and low motivation. In the experiments, crowds (80–100 participants) were asked to imagine that they were entering a concert hall consisting of a narrow bottleneck. The low motivation group reported feeling bored during the interruption, while the high motivation group reported feeling impatient. Additionally, a motivational decrease was observed for the high motivation group due to the interruption. This drop in motivation after the interruption is also reflected in the measured density (person/m$^2$) in front of the bottleneck. Overall, both studies showed that interruption can have significant effects on individuals, including psychological and physiological impacts. The observed motivational decrease through interruption is particularly relevant for crowd management, but further investigation is needed to understand the context-specific effects of interruptions.

interruption_gameplay - DOI: 10.34735/ped.2021.
11 Study 2: https://ped.fz-juelich.de/da/doku.php?
id=interruption_crowd - DOI: 10.34735/ped.2021.
12.

**Funding:** The authors received no specific funding
for Study 1. Study 2 was part of the project "CroMa
—Crowd Management in Verkehrsinfrastrukturen."
The project was funded by the German Federal
Ministry of Education and Research
(Bundesministerium für Bildung und Forschung)
under grant number FKZ 13N14531. The funders
had no role in the design of the study; in the
collection, analyses, or interpretation of data; in the
writing of the manuscript, or in the decision to
publish the results.

**Competing interests:** The authors have declared
that no competing interests exist.

# General introduction

Being interrupted is one of the most common incidents in our daily lives. The natural order of
the world, as well as many actions of individuals or groups, intersect at some level, making it
improbable that these would never clash. A person can get interrupted while thinking, focus-
ing, deciding, acting, or trying to finish something. This interruption can occur via a sudden
noise, a message on the phone, a power cut, an earthquake, or just a random shift in attention.
There are countless things that can interrupt people, and human brains are trained to cope
with these surprise encounters to some extent [1].

Because it is so common, researchers have extensively studied and conceptualized interrup-
tion throughout the history of psychology. Interruption has been elaborated within various
contexts such as memory [2–4], recovery of attention [5–7], learning [8], emotion [9], motiva-
tion [10], organizational behavior [11,12], and so on. Some notable studies have found that
interruption promotes better recall [2], creates a tendency or urge to return to the unfinished
task [13], produces emotional behaviors [9,14,15], can evoke anxiety [12,14,16], or even spark
positive feelings [12,17].

Pedestrian events can be seen as a prominent context for interruptions to occur. The modern
world has complex and effective transportation and pedestrian systems with many modes of
transport (trains, cars, planes, sidewalks, etc.), however, the cost of having multimodal traffic is
that people must wait for the intersecting objectives and routes of other travelers from time to
time. This situation is heavily normalized since people expect a 'pause' to some extent and have
a higher tolerance for the event (e.g., waiting at a stoplight, train delay). However, interruption
also has a "surprise" component, which can be seen in many studies as mentioned above [1].
Even though it is normal for pedestrians to wait for a stoplight to some degree, some stoplights
do not turn green quickly depending on the road and intersection density, or one might
encounter more stoplights than expected (or no crosswalks) due to insufficient infrastructure.

So far, few studies have addressed interruption within the field of pedestrian dynamics.
Tang, Huang, and Shang [18] added interruption (i.e., ticket control) as a component to their
pedestrian-following simulation of plane boarding to acquire a more realistic model. Khan
and Hoque [19] discussed the topic in a vehicle traffic context and argued that smooth traffic
could be achieved with fewer interruption. Chen and Wang [20] investigated whether unex-
pected pedestrian crossings, as an interruption to vehicle flow, are a significant factor for over-
all traffic. Unfortunately, studies on the topic of pedestrian interruptions are far too few, and
so far, the focus has solely been on the "flow disruption" of either pedestrians or vehicles. How-
ever, to have an overarching understanding of the subject, the psychological perspective of
interrupted pedestrians must be investigated, and various types of interruption must be
explored. This paper aims to provide a basis for a discourse on the above issues. A motivational
framework, along with basic psychophysiological responses (heart rate) and crowd dynamics
parameters (density), was chosen for investigating how different conditions result in different
reactions to the interruption of pedestrian flow.

From the perspective of motivational theory, the phase of moving toward a goal is called
the volitional process. The implementation of goal pursuit (volitional process) is set in motion
as the post-decisional action, after the expectancy and value arrangement of the motivational
state of mind [3]. An anticipated pause, such as waiting momentarily at the stop lights, would
not affect the person since coping with what is expected is part of the process. However, if peo-
ple do not expect to spend time waiting for more than usual, an interruption can alter the voli-
tional process and potentially produce psychophysiological and psychological reactions.

Previous psychology studies with similar contexts investigated interruption and found var-
ied results. It has been found that people may feel irritated and describe the situation as

frustrating and unfair [21]. The situation can alter a person's perception and subjectively increase the delay time, making it seem longer than it actually is [21–23]. People can experience an impatience state [23], boredom [24], or anger [25] depending on the situation of the interruption event and the psychological background of the person (i.e., personality traits). Previous findings regarding psychophysiological reactions are also illuminating for exploring interruption in a pedestrian context. Studies with similar aims found that interruption causes an increase in heart rate and skin conductance levels, and has detrimental effects on working memory after the interruption [26,27].

So, what makes these reactions vary in different interruption events during pedestrian activities? One explanation could lie within the 'valuation' concept, which implicitly increases the given value for the goal as time passes, that is, as the person is getting closer to the goal [23,28]. A high valuation could potentially result in a more negative and aroused reaction when there is an interruption, whereas a low valuation could result in a more neutral reaction. Study 1 will focus on this distinction and formulate interruption as 'early' and 'late,' with different motivational and emotional responses, respectively.

Another possible explanation for different reactions to interruption could be a simple motivational distinction, that is, whether a person's motivation is high or low from the beginning. High or low motivation is known to alter attention, performance, and learning differently [29]. The starting motivation could likely lead to different outcomes if maintained until the interruption and could potentially alter the motivation after the interruption. Study 2 will, therefore, investigate and explore this notion while distinguishing between having 'high' and 'low' motivation.

## Study 1

### Introduction

'Valuation' is a concept whose potential to effect different interruption outcomes seems apparent since goal proximity is directly associated with the timing of the interruption. The goal proximity concept states that when people are close to their goals, then the valuation of the goal is much higher when compared with others who are still at the beginning of their goal pursuit [23]. In other words, being closer to an end (perception-wise, lengthwise, timewise) for achieving a goal increases the attractiveness of the task and its outcome.

When people are attracted to their goals, an interruption is potentially more disrupting and produces more annoyance when compared to situations in which people have low or no goal attraction [30,31]. It can be assumed that a high physiological and psychological arousal would emerge within people when they are interrupted at the end of their goal pursuit. In contrast, people would be in a state of low arousal when they are interrupted at the beginning of the goal action.

What exactly do people feel in these kinds of situations? Motivation and state emotion literature provide some hints for structuring the different states that people experience in these kinds of situations. Although it has been studied in many different contexts, arousal theories define "state boredom" as unwanted arousal when there is a discrepancy between the task and the environment [32]. This discrepancy can easily be caused by an interruption event since the task cannot be completed due to the environment. Additionally, most studies have divided state boredom into high arousal and low arousal boredom (arousal in both physiological and psychological terms) [32,33]. So far, studies have not agreed on a universal explanation for why boredom varies so much while it manifests itself physiologically and psychologically [32]. High arousal boredom has components of agitation with a higher heart rate, connected to a

negative feeling, and an urge to flee from that state, while low arousal boredom is more related to a lethargic experience and tiredness with a lower heart rate [34].

While low arousal boredom can be considered as a more 'conventional' boredom, high arousal boredom has an 'impatience' component, and some studies have described state impatience similarly to what has been described for high arousal boredom [35]. Unfortunately, state impatience (or volitional impatience) that occurs in the post-decisional process has received little attention in the literature so far and does not include detailed theorizing, but it can still be used for labeling the state to create a more basic contrast with boredom. Most importantly, it can be assumed that people are expected to perceive a subjective time pressure, which is a key component of state impatience [35], when they are highly goal-focused, and an interruption occurs.

In this study, instead of using predefined state boredom components, impatience and boredom concepts were used separately, with their respective definitions in the state boredom discourse. Hypotheses for Study 1 were formed based on the expected psychological and physiological outcomes of the interruption state. An early interruption would produce a low arousal state, which would consequently be labeled as boredom. A low arousal state consists of a higher score on the disengagement subscale of state boredom and a lower heart rate. On the other hand, being interrupted while being nearly at the end would produce a highly aroused state due to the high valuation of the goal, and this state would thus be considered impatience. A high arousal state consists of a higher score on the high arousal subscale of state boredom and a higher cardiac output.

Due to Covid-19 restrictions during the study period, it was not feasible to conduct experiments with multiple participants in a pedestrian context. Thus, the study design was modified to solely focus on a game-playing context, involving one participant at a time. The study was designed to analyze how interruptions affected participants' emotions and heart rate levels while they were engaged in a video game.

## Method

**Participants and design.** 61 participants between the ages of 18 and 50 (mean age = 24.5; 41% female, 59% male) were recruited from Germany (18) and Turkey (43). The experiments were conducted in two countries due to practical reasons: participant collection in Germany during May and June 2021 was not successful enough to complete half of the quota due to the Covid-19 pandemic. The remaining participants were recruited in Turkey during July and August 2021, when/where the Covid-19 regulations were not too strict. Both samples had an equal number of participants distributed between experimental conditions.

After receiving a written ethical approval from the ethical committee of Bergische Universität Wuppertal, participants were recruited through Facebook posts, university announcements, physical posters, or verbal calls among acquaintances. All participants were provided with explanation links of the procedure and the payment (20 euros) they would receive for a two-hour experiment in which they would play a computer game.

The study design was a between-subject design; hence, all participants were randomly assigned to a condition (early interruption = 31 or late interruption = 30) without their knowledge of which condition they would take part in or whether there would be any condition at all. The true nature of the study was withheld from the participants until the completion of their respective experiments.

**Measures.** *Scales*. Two factors of the *Multidimensional State Boredom Scale* [32] were used to assess the overall state of participants. The scale consists of five factors: Disengagement, high arousal, low arousal, inattention, and time perception. Only the first two factors

(disengagement and high arousal; 10 and 5 items, respectively) were used in this study as a two-dimensional measurement. Half of the items in the disengagement subscale were not used due to their association toward a broader time period, rather than focusing solely on the situation or the event (e.g., "*Everything seems repetitive and routine to me*"). Consequently, a total of 10 items (5 items from each factor) were used in this study to assess the state emotion of the participants. Throughout the article, these subscales will deliberately be written as "boredom" and "impatience" to avoid confusion with the other measurement methods. The response format was a 7-point Likert scale from 1 = *Strongly disagree* to 7 = *Strongly agree*, and higher scores indicated higher levels of the boredom/impatience-related state. These items were hidden and scattered among self-made deceptive items. Approximately 30 items were created to hide the main items so that participants would not be suspicious that the interruption was intentional. The deceptive items varied in context and included happiness, excitement, anger, etc.

*EcgMove 4 –ECG and activity sensor*. Heart rate data was collected using an EcgMove 4 [36] device during the participants' gameplay and interruption periods. The device included two electrodes and was presented to participants as an easy-to-wear product that they could attach to their lower-left chest by themselves. Participants wore the device before the experiment began, and the recording started before they attached the device and stopped when the data was being exported.

As the device was unable to start and stop the recording remotely, we cut the raw data of each participant into their respective time periods (interruption, game playing, baseline) using Unisens Viewer software [37]. We used the exact time points that were collected during the experiment (i.e., start of the interruption and end of the interruption) for this purpose. Afterward, we calculated beats-per-minute averages of participants for the cut periods individually with Unisens Analyzer software [37]. The numerical data was then stored as an xls file for hypothesis testing in SPSS.

Due to the distortions and artifacts within the heart rate data, approximately one-third of the data was excluded from the main dataset. The main reason for this was that the experiments were held in the summer season, and it was up to 40 degrees Celsius in Turkey (average outdoor temperature for the experiment days = 36.2˚C) at the time. The experiments were conducted in private offices with air conditioning, but it was not enough to create environment equivalent to that of Germany (average outdoor temperature for the experiment days = 24.4˚C) in terms of temperature. For some cases, the heart rate device could not correctly gather the data due to the participants sweating before they arrived at the experiment location. The remaining data were processed as explained above.

**Procedure.**   Each participant took part in the experiment individually. The time slot was decided as two hours for each participant since the experiment could have potentially lasted for more than one and a half hours. The approximate finishing time among participants was around one hour. The true nature of the experiment was withheld from the participants until the debriefing at the end.

Upon arrival, participants were informed that they would be playing a computer game [38] while wearing a heart rate device to measure their excitement level. They were also instructed to fill out a questionnaire about their overall experience at the end of the game. The experimenter explained that they would be monitoring the participants' game play via a Zoom screen share to record the exact times they entered specific locations. The game involved finding a specific place and playing hide-and-seek with a talking rock. The participants needed to find the talking rock three times to complete the experiment.

In reality, the experimenter only observed the participants to monitor their overall progress. In the "early interruption" condition, the experimenter interrupted the participants 15 minutes

into their gameplay session by knocking and opening the door of the experiment room. The participants were informed that they needed to stop playing due to a technical problem with the heart rate device that needed to be addressed, and they were instructed to wait for "a little while". In the "late interruption" condition, the process was similar, but there were no set time markers; the interruption time was decided when the participants were close to completing the given task (i.e., finding the rock twice). The late interruption time points varied from half an hour to one and a half hours. Interruptions lasted exactly ten minutes in each condition.

During this ten-minute interruption, participants were asked to wait without doing anything. They were not allowed to play the game, go out of the room, check their phones, or use the computer in front of them. After the interruption finished, the experimenter came to the room again and acted as if the problem was solved but stated that the situation was also interesting for them. The experimenter recommended that the questionnaires should be filled out now instead of after finishing the game. While filling out the questionnaires, participants were asked to focus on the interruption experience rather than the actual game experience. After finishing the questionnaires, participants were asked to choose whether they wanted to continue playing or stop the experiment.

After quitting the game, participants were briefed about the true nature of the study and the reasons for the deception. They were told that they could withhold their consent if they wished to. Participants were also asked whether they had realized what was happening. Around ten participants (15%) stated that they were suspicious, but they had no idea about the exact nature of the study. Their data was also considered valid since they did not suspect the actual reason for the interruption.

## Results

**Boredom and impatience.** To explore whether early and late interruption produced different emotional/motivational states, two independent samples t-tests were conducted. The first analysis tested whether early interruption produced a boredom state more than late interruption. The analysis showed that there was no significant difference between early (M = 4.05, SD = 1.3) and late (M = 4.02, SD = 1.19) interruption in terms of boredom; $t(59) = .08$, $p = .94$, suggesting that the predicted increasing effect of early interruption on boredom state was not found. The following analysis was conducted to analyze whether late interruption produced an impatience state in the participants more than early interruption. The analysis also showed that there was no significant difference between late (M = 3.56, SD = 1.28) and early (M = 3.72, SD = 1.38) interruption in terms of impatience; $t(59) = .47$, $p = .64$.

**Heart rate.** Mean beats per minute (BPM) were measured for every participant to explore the heart rate differences across variables. The mean of the interruption period (approximately 8 minutes; M = 84.28, SD = 11.31) and game-playing period (approximately 8 minutes; M = 82.72, SD = 11.84) were calculated for analyzing the main hypotheses. Additionally, a baseline period (questionnaire filling after the interruption period; approximately 8 minutes; M = 82.56, SD = 10.66) was calculated to see whether both game playing and interruption periods differed from a low arousal-inducing activity.

A repeated measures ANOVA showed that mean BPM differed significantly between different time periods regardless of whether the interruption period was early or late; $F(2, 72) = 4.71$, $p = .012$, $\eta p2 = .12$ (Fig 1). The post hoc Bonferroni corrections revealed that there was a significant difference between the baseline and interruption periods ($p = .013$), but the difference between the baseline and the game-playing periods was not significant ($p = 1$). However, follow-up paired samples t-tests showed a significant difference between the mean BPM of overall playing and interruption periods, $t(42) = -2.65$, $p = .011$. The results indicate that the

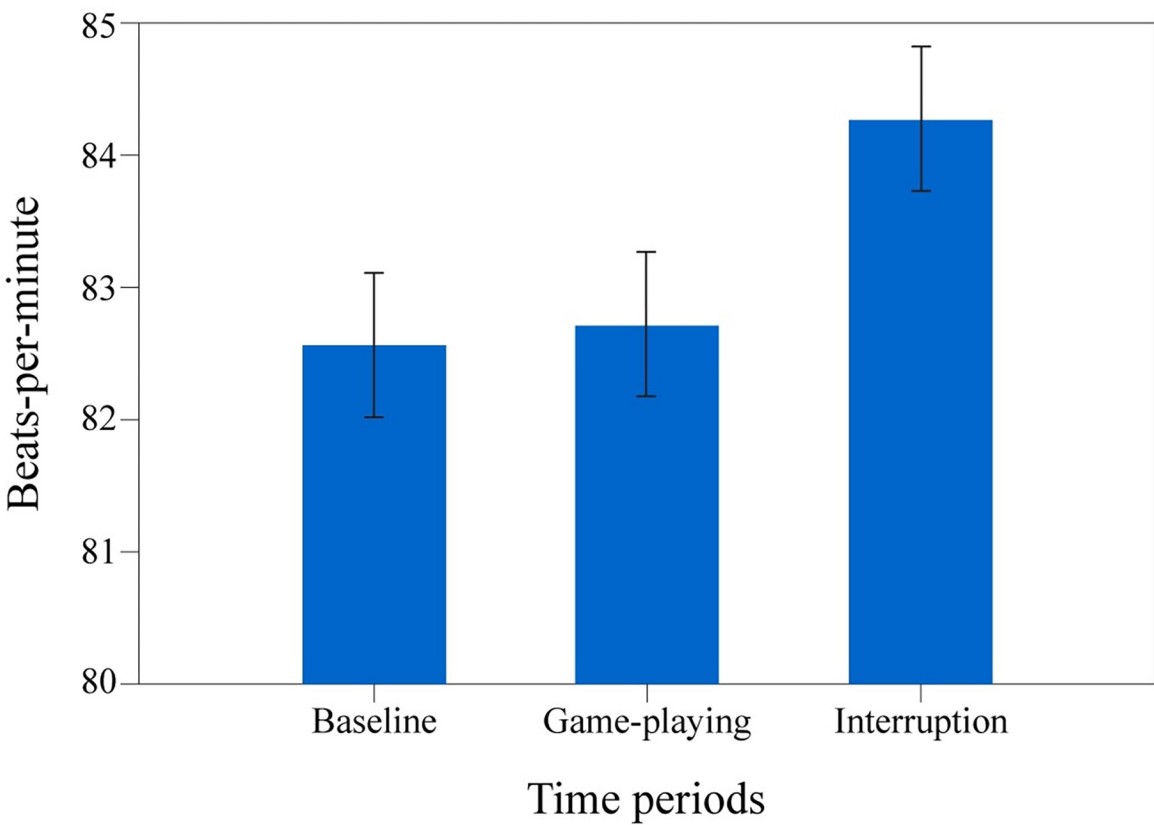

**Fig 1. Mean values of participants' heart rate.** Baseline, game playing, and interruption time periods.

game-playing period does not differ from the baseline in terms of heart rate and can be described as a "normal" activity. Consequently, the interruption period significantly differs from both and can be interpreted as an "arousal-inducing" activity.

Paired and independent samples t-tests, along with ANOVAs, were conducted to explore whether early and late interruptions produce different outcomes in terms of heart rate. Focusing only on the interruption period, the analysis of the mean BPM of the interruption time between the early interruption group (M = 85.7, SD = 9.77) and the late interruption group (M = 81.52, SD = 12.1) showed that there was no significant difference between the conditions, $t(45) = -1.3$, $p = .2$, indicating that the interruption produces approximately the same effect in both conditions.

A 2 (BPM: playing & interruption) x 2 (condition: early & late interruption) mixed-design ANOVA (see Fig 2) revealed a significant main effect of the BPM period, $F(1, 41) = 6.88$, $p = .012$, $\eta p2 = .14$. As previously stated, the interruption period had a significantly higher BPM than the game-playing period. However, the main effect of interruption type, $F(1, 41) = .88$, $p = .35$, $\eta p2 = .02$, and the interaction between BPM period and interruption type, $F(1, 41) = .02$, $p = .89$, $\eta p2 = .00$, were non-significant. Although the analysis was not significant, Fig 2 suggests that BPM was generally higher in the early interruption than in the late interruption condition. The overall course of the experiment can potentially explain this situation: the heart rate data for all participants showed a decreasing trend throughout the experiment period. Still, the effect is plausible since the arousal can often arise in a vague or ambiguous situation [39], such as participating in a psychology experiment. After time passes, the arousal can fade throughout the period if the event begins to be perceived as normal.

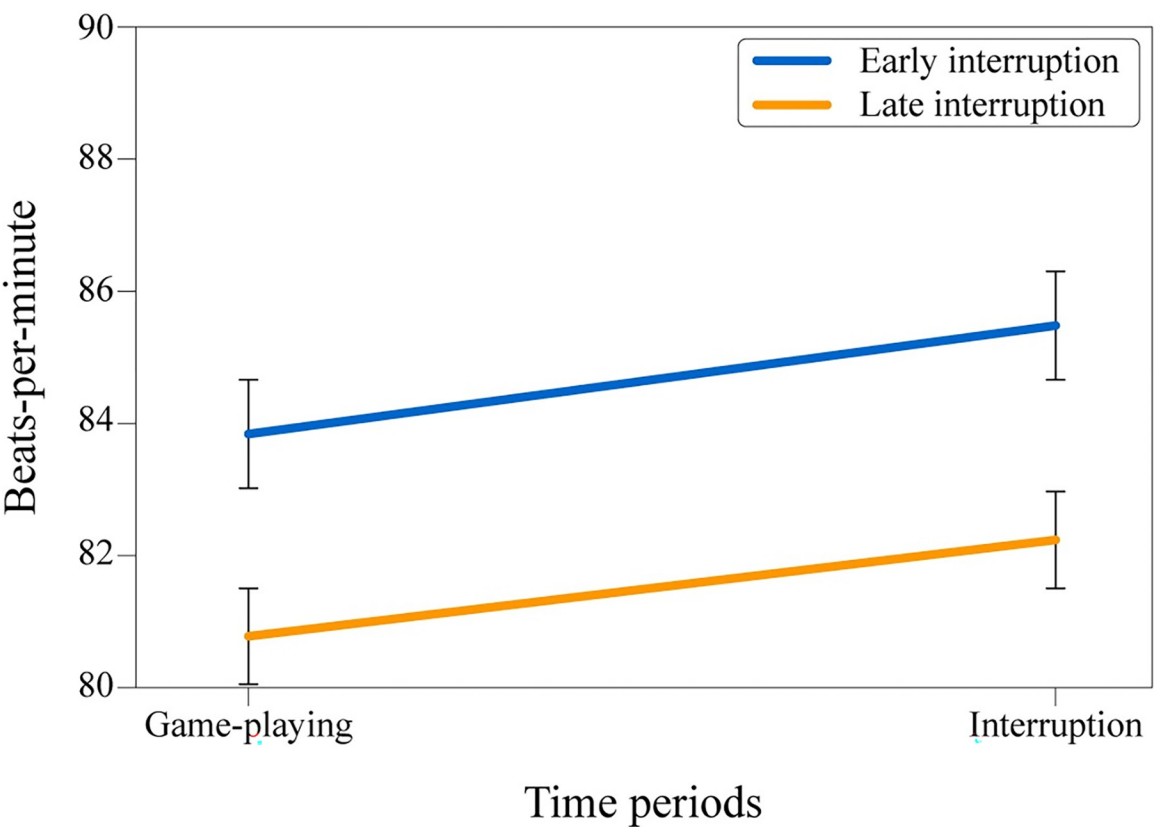

**Fig 2. Mean values of participants' heart rate.** Game playing and interruption time periods across conditions.

Lastly, samples from Turkey and Germany were analyzed with independent samples t-tests to observe whether these two groups differed. Neither the interruption period nor the game-playing period showed a significant difference between participants from Turkey and Germany, $t(45) = -1.3$, $p = .19$, and $t(47) = -1.2$, $p = .87$, respectively.

## Discussion

Study 1 investigated whether interruption time, as early and late interruption, has a differentiating effect on the perceived emotional state and physiological arousal of people. It was hypothesized that early interruption leads to a 'bored' state and late interruption leads to an 'impatient' state, both psychologically and physiologically. The collected data showed no evidence for these hypotheses.

According to the results of the boredom and impatience questionnaire items, it does not appear that people associated their respective states with boredom when they were interrupted right after they started their goal pursuit, nor did perceived impatience increase when people were interrupted in the later stages of their goal pursuit. Furthermore, heart rate data showed an overall increase during the interruption period. The expectation was that a higher increase would be observed in the late interruption group than in the early interruption group. However, the increase was only valid for the whole sample but not for the two experimental conditions. Different location samples (Germany and Turkey), gender, or other demographic variables also showed no difference between their respective groups in any means of data collection.

To summarize, the timing of the interruption does not seem to have an effect on producing different negative state emotions. However, interruption itself led to an overall increase in perceived annoyance, both physiologically and psychologically, and there was no difference in annoyance levels between early and late interruptions.

## Study 2

### Introduction

Study 2 focuses on interruption outcomes in a crowd context. Due to the ease of Covid-19 restrictions, it was possible to have a 'crowd' experiment by taking into consideration the adjusted health rules. Study 2 had a bottleneck setup in which 80 to 100 participants were instructed to rush towards and through a gate to enter an imaginary concert (for six consecutive runs). Additionally, two levels of a motivational drive were used in this study: High and low motivation for reaching the bottleneck. Previous studies have shown that high or low motivation produces distinct outcomes in attention, performance, and learning [29]. Furthermore, motivation has proven to be one of the key factors in crowd dynamics [40]. It is expected that the experience of an interruption depends on motivation and is more disruptive in a goal pursuit while highly motivated when compared with low motivation. Psychological and physical arousal during an interruption is expected to be higher in people with high motivation and lower in people with low motivation. Unlike Study 1, which used "timing" (early & late) as an interruption component, Study 2 used instructed motivation (low & high) instead. Impatience and boredom factors were also assumed to be significant for this setup since the definition of state boredom as "unwanted arousal caused by task and environment discrepancy" is still heavily related to the experimental setting.

The hypotheses for Study 2 were formulated based on the same framework as Study 1. It was hypothesized that an interruption for people with low motivation would result in a low aroused state, and this state can consequently be labeled as boredom. A low aroused state consists of a higher point in the disengagement subscale of state boredom and a lower heart rate. In contrast, an interruption for people with high motivation would lead to a highly aroused state, and this state can be considered impatience. A high aroused state consists of a higher point in the high arousal subscale of state boredom and higher cardiac output. However, due to the movement factor of the experiment, an increase in heart rate during the interruption period compared to the walking phase was not hypothesized. As people would not be able to move forward during the interruption period, a "constantness" for the heart rate was hypothesized instead of an increase. Details regarding this aspect will be discussed in the next sections. An additional questionnaire item was created for people with high motivation, and it was hypothesized that an interruption for highly motivated people would reduce their level of motivation.

### Method

**Participants and design.**   The experiments in Study 2 were part of a larger set of experiments in that included various crowd studies conducted over four days, with a total of 1200 participants [41]. The experiments were conducted at Mitsubishi Electric Halle in Düsseldorf, a concert venue, in October 2021. All participants were recruited through local newspapers and media announcements, and they were briefed about the experiments and any risks involved. Participants signed informed consent forms before participating, and a written ethical approval for the experiments had been obtained from the ethical committee of Bergische Universität Wuppertal prior to conducting the experiments. Each participant received 70 euros per day (between 10 a.m. and 4 p.m.). The study experiment was held on the third day

after the lunch break. Three groups, each consisting of 80–110 participants, were randomly selected for the interruption experiments, resulting in a total of 301 participants. The age of the selected participants ranged from 18 and 75 (Mean age = 35,6). The gender distribution among participants consisted of 154 (51%) women, 128 (42%) men and 19 (7%) others.

The study design for Study 2 was a mixed design, where each group participated in the study twice (three groups, a total of six runs); first without interruption and then with interruption. Once again, participants were not informed about the true nature of the study at the beginning, as the interruption event was not mentioned beforehand. The three groups were divided based on their assigned condition, with two groups having high motivation and one group having low motivation. The assigned motivation condition was used for both runs, with and without interruption.

**Measures.**    *Scales*. Two factors (disengagement and high arousal; a total of 10 items) of the Multidimensional State Boredom Scale [32] were used to assess the overall state of participants within an interruption context as in Study 1. The response format was a 7-point Likert scale ranging from 1 = Strongly disagree to 7 = Strongly agree, with higher scores indicating higher levels of the boredom/impatience-related state.

The *Motivation Scale*, created for CroMa-Project, was also used to measure how people felt during a bottleneck/interruption situation. After their experimental run had finished, the scale was presented to participants twice, and they were asked to evaluate two different situations: one considering the time period before the interruption and another considering the time period after the interruption. The aim was to measure the change in motivation of the participants depending on the interruption.

Due to time constrains and the rapid structure of the experiments in Study 2, no deceptive questionnaire items were employed. These items were exclusively administered to participants in Study 1 to conceal the primary objective. However, in Study 2, presenting these items (which comprised more than 60% of the overall items) was deemed redundant and time-consuming, given that interruptions are inherent in pedestrian experiences. Following the questionnaire, participants were debriefed on the true nature of the study.

*EcgMove 4 –ECG and activity sensor*. 20 EcgMove 4 [36] devices (maximum number of devices at our disposal) were used to collect HR and HRV data from participants during the experiments. 20 participants from each group (a total of 60) were randomly selected and instructed to wear the device before the experiment began.

The raw data of each participant was then cut manually into their respective time periods (interruption, before interruption, and after interruption) using Unisens Viewer software [37]. These time periods, along with their start and end time points, were recorded during the experiments (as in Study 1). The recorded time points were based on the entire group's action: The before interruption time period started when participants were instructed to go to the bottleneck area and ended when they were interrupted; the interruption time period ended when they were informed that they could proceed; the after interruption time period ended when the last person went through the bottleneck.

After the cutting process, beats-per-minute averages of participants for the cut periods were calculated individually with Unisens Analyzer software [37]. The numerical data was then stored as an xls file, for hypothesis testing in SPSS.

*Feedback terminal*. Participants were presented with a smiley feedback terminal [42] as they exited the bottleneck area. They were encouraged to tap one of the smiley buttons on the device to provide feedback on their overall experience of the experiment. The terminal displayed the question "How did you feel in the experiment?" and included four different smileys, namely, very happy (4), happy (3), unhappy (2), and very unhappy (1). Almost all participants

clicked on the device in each experimental run. The data was then stored as an xls file for testing in SPSS.

*Video recordings.* Video and audio recordings were taken from three top-down cameras (focusing on different parts of the experiment area) to capture the trajectory data of pedestrians. These trajectories were later used to calculate the density of each group in different time periods. The procedure was as follows: PeTrack software [43] was used to detect the trajectory paths of all pedestrians. All trajectories were individually checked and corrected using PeTrack. Manually corrected trajectories from the three cameras were then merged into one full trajectory file for each experimental run. These files were in.txt format and contained pixel coordinates for everyone for each frame. These files were then used to calculate individual Voronoi densities and plot them using the PedPy package in Python [44].

Video recordings were also used in this study to gather the exact timing of the heart rate recordings and for observational purposes. Participants were briefed about the recordings prior to the experiment.

**Procedure.**  Study 2 had a bottleneck/pedestrian context and was designed to withhold the interruption information from participants. Experimenters acted again as if a technical problem caused the interruption.

Participants were asked to imagine a context where they are about to watch their favorite singer in a concert. In the high motivation condition, participants were told the following: "Imagine you are on your way to a concert by your favorite artist. You know that at the back you can hardly see anything at all or only the video screen. You absolutely want to be standing next to the stage and therefore want to access the concert as fast as possible. After a signal, we will open the entrance" (translated from German). In the low motivation condition, they were instructed: "Please imagine that you are on your way to a concert by your favorite artist. You know that everyone will have a good view. Still, you would like to access the concert quickly" (translated from German).

After these instructions, each group walked to the area directly in front of the bottleneck and waited for the bottleneck to be opened. The first run for each group was always a "without interruption" run: Participants were able to cross the platform after the bottleneck was opened. In the "with interruption" runs, experimenters interrupted the participants after a couple of seconds with a verbal 'stop' order while acting as if there was something wrong with their technical equipment. The interruption lasted approximately two minutes, and participants were instructed to wait and not cross the bottleneck during this period while remaining in their position (see Fig 3 for the complete process).

The first group was the low motivation group. The second and third groups were high motivation groups because a repetition was thought to be needed; a person fell to the floor in the second group during the run, although it was later decided to use the data of both high motivation groups.

After each run, participants were directed to tap the feedback terminal and were instructed to fill out a questionnaire. Afterward, participants were briefed about the true nature of the experiment.

## Results

**Feedback terminal.**  A two-way ANOVA was conducted to examine the effect of interruption (without [M = 3.24, SD = 0.94] and with [M = 2.71, SD = 1.02]) and the motivational condition (high [M = 2.45, SD = 1.02] and low [M = 3.49, SD = 0.72] motivation) on the overall mood of the participants. There was no significant interaction effect ($F(1, 588) = 3.156$, $p =$ .076, $\eta p2 = .001$, see Fig 4) but there were two main effects: A significant difference in the

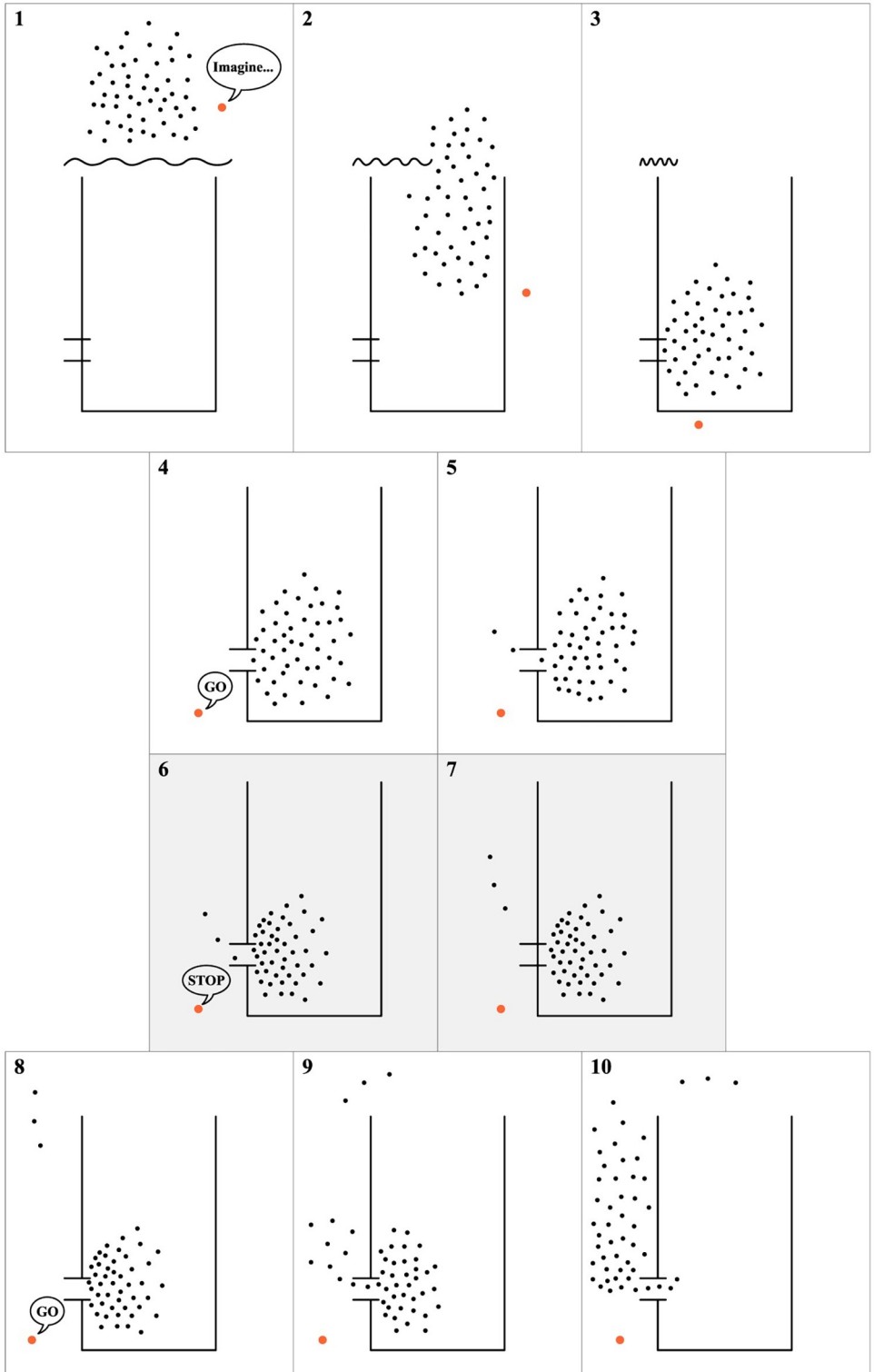

**Fig 3. Complete experimental process.** The order of the drawings can be followed through the image numbers. The grayed background drawings represent the interruption period (two minutes). The orange dot represents the position of the experimenter giving instructions.

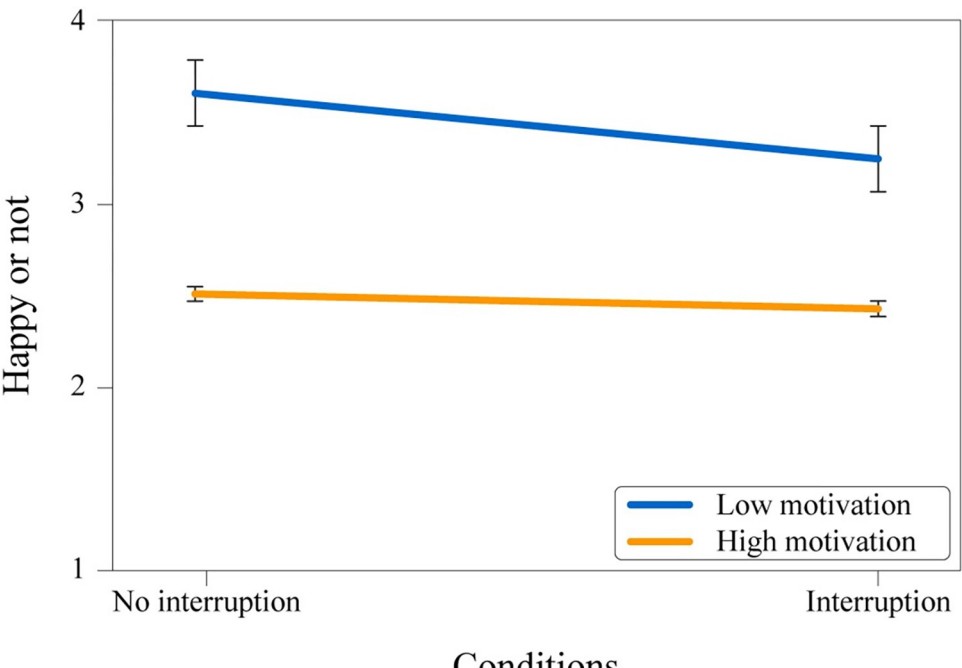

**Fig 4. Mean values of participants' mood on interruption (with/without) and experimental condition.** 1: Very unhappy, 4: Very happy.

mood of the participants between motivation groups ($p < .001$), and between runs with and without interruption ($p = .004$). These results indicate that the mood was better during low motivation runs than the high motivation runs and better during runs without interruption compared to runs with interruption.

**Boredom and impatience.** To explore whether having high and low motivation produces different emotional states, two different independent samples t-tests were conducted for the interruption runs. The first analysis aimed to test whether having low motivation during an interruption incident produces a greater sense of boredom in participants than having high motivation. The analysis showed a significant difference between low motivation (M = 4.22, SD = 1.48) and high motivation (M = 3.82, SD = 1.56) in terms of the participants' perceived boredom state; $t(293) = 2.12$, $p = .035$, indicating that an effect of low motivation on boredom perception.

The second analysis was conducted to see if having high motivation produces an impatience state in the participants more than having low motivation during the interruption. The analysis confirmed that there was a significant difference between having high motivation (M = 3.37, SD = 1.55) and having low motivation (M = 2.69, SD = 1.46) for the impatience state; $t(296) = 3.69$, $p < .001$. These results correspond to the initial hypothesis.

Follow-up tests were conducted for all runs (without interruption and interruption runs combined) to explore solely impatience and boredom items (I feel bored & I am impatient right now) because only these items were presented to the participants in without interruption runs instead of the whole scales. A two-way ANOVA was conducted to examine the effect of interruption and instructed motivation on the "I feel bored item." There was no significant interaction between the dependent variables on this item; $F(1, 596) = .38$, $p = .54$, $\eta p2 = .001$. However, simple main effects analyses showed that there was a significant difference in feeling bored between motivation groups ($p = .001$), and there was a significant difference in feeling

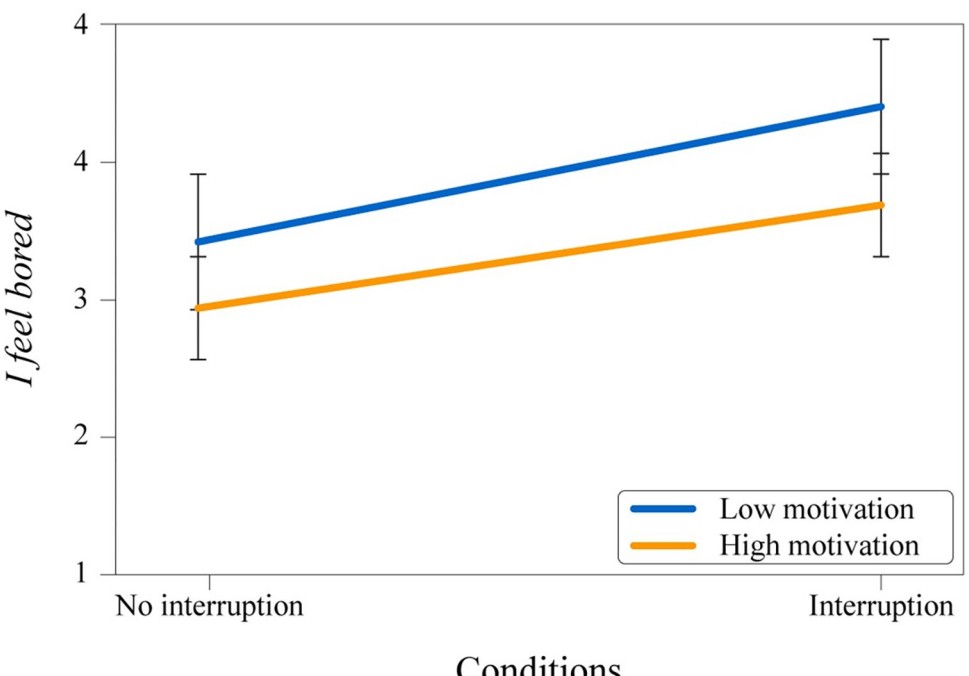

**Fig 5. Mean values of "I feel bored" item.** Interruption (with/without) and experimental condition.

bored between the runs with and without an interruption ($p < .001$) (see Fig 5). Another two-way ANOVA was conducted to examine the effect of interruption and instructed motivation on the "I am impatient right now item." There was a significant interaction between the dependent variables on this item; $F(1, 601) = 4.3$, $p = .039$, ηp2 = .007, indicating the impact of being interrupted or not on impatience depends on the initial motivation level (see Fig 6). Simple main effects analyses also showed that there were significant differences in being impatient between motivation groups ($p < .001$), as well as between whether interruption happened or not ($p = .038$).

**Motivation.** A 2 (motivation level: motivation before the interruption & motivation after the interruption) x 2 (condition: high or low motivation instruction) mixed-design ANOVA (see Fig 7) was conducted to observe whether motivation levels decreased due to the interruption event. The analysis showed that there was no significant main effect of the interruption on motivation level; $F(1, 589) = 3.34$, $p = .068$, ηp2 = .006. However, the main effect of the initial motivational instruction was significant, $F(1, 589) = 79.67$, $p < .001$, ηp2 = .119, along with the interaction between the initial motivational instruction and interruption, $F(1, 589) = 24.03$, $p < .001$, ηp2 = .039. Results indicated that the interruption decreases the motivation level when people are highly motivated but it has the opposite effect when people have low motivation.

**Heart rate.** Mean beats per minute (BPM) were measured for 20 participants in each group for a total of 60 participants (high motivation, M = 90.16, SD = 12.25; low motivation, M = 85.09, SD = 16.07) to explore the heart rate differences across variables. The mean of interruption period (approximately 2 minutes; M = 82.9, SD = 15.17), the mean of "before interruption" period (approximately 2 minutes; M = 92.14, SD = 15.98), and the mean of "after interruption" period (approximately 2 minutes; M = 91.61, SD = 13.36) were calculated for analyzing the main hypotheses for the interruption runs. Due to the distortions and artifacts within the heart rate data, the data of five participants were excluded from the dataset.

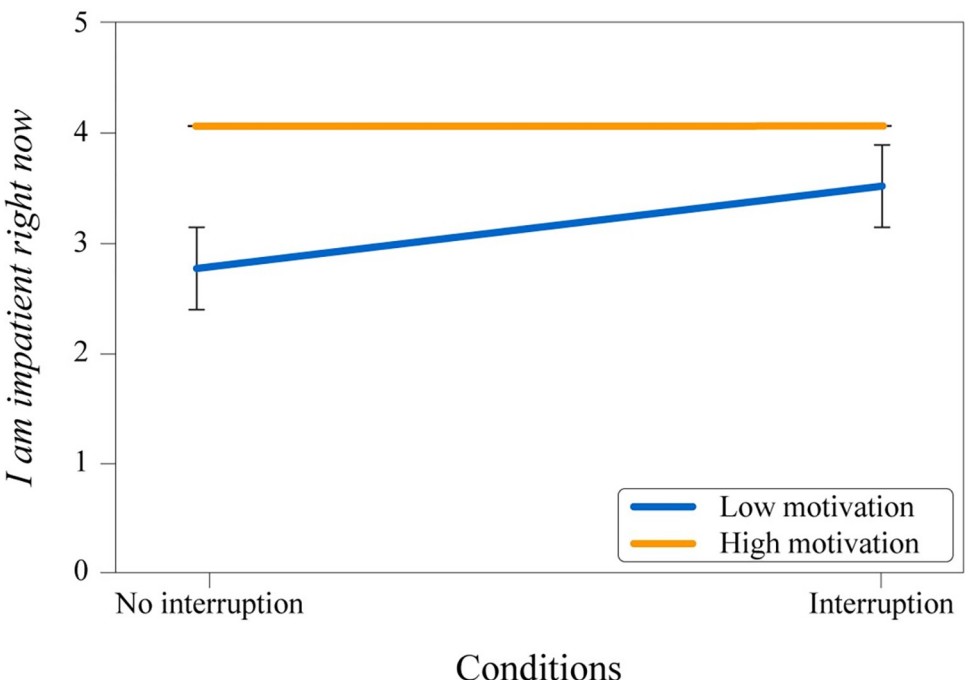

**Fig 6. Mean values of "I am impatient right now" item.** Interruption (with/without) and experimental condition.

A repeated measures ANOVA showed that mean BPM differed significantly between different time periods in interruption runs regardless of the instructed motivation; $F(2, 90) = 30.21$, $p < .001$, $\eta p2 = .401$. The post hoc Bonferroni corrections revealed that there was a significant difference between before interruption and interruption periods ($p < .001$) and after interruption and interruption periods ($p < .001$): The BPM was lowest during the interruption period. However, there was no significant difference between before and after interruption periods ($p = 1$). Therefore, the expected increase or constancy between the interruption period and the other periods was not found; heart rate data showed a significant decrease during the interruption period. The discussion section will explore possible reasons for the unexpected 'opposite direction' effect.

An independent samples t-test was conducted to explore whether having high or low motivation matters regarding the participants' heart rate. Focusing only on the interruption period, the analysis of the mean BPM between high (M = 84.05, SD = 14.52) and low (M = 80.53, SD = 16.68) motivation conditions showed that there was no significant difference between conditions, $t(47) = .758$, $p = .45$. The results showed that interruption produces the same effect in terms of heart rate in both conditions. Still, the relatively small sample size might have caused this outcome.

A 3 (BPM periods: before & during interruption & after) x 2 (condition: low & high motivation) mixed-design ANOVA (see Fig 8) revealed that there was a significant main effect of the BPM period, $F(2, 88) = 24.763$, $p < .001$, $\eta p2 = .36$, indicating that periods had significantly different heart rate outputs. However, the main effect of motivation type, $F(1, 44) = 1.43$, $p = .238$, $\eta p2 = .032$, and the interaction between BPM period and motivation type, $F(2, 88) = 1.89$, $p = .157$, $\eta p2 = .041$, were not significant (potentially caused by the relatively small sample size). The results suggested that the effect of motivation type over heart rate was not found.

Lastly, a 2 (BPM periods: before interruption period & after interruption period) x 2 (condition: low & high motivation) mixed-design ANOVA (see Fig 9) was conducted to explore hypothesized motivational decrease across conditions in the heart rate context. The analysis

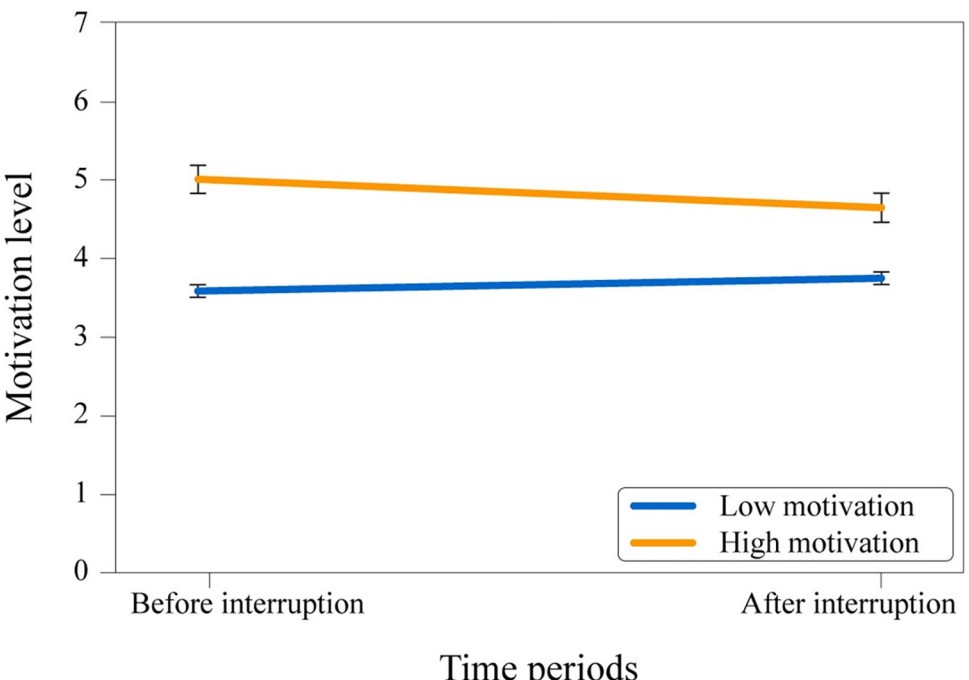

**Fig 7. Mean values of different motivation time periods and experimental conditions.**

revealed that the main effect of the BPM period; $F(1, 44) = .03$, $p = .86$, $\eta p2 = .001$, the main effect of the motivation type; $F(1, 44) = 2.23$, $p = .143$, $\eta p2 = .048$, and the interaction between BPM period and motivation type; $F(1, 44) = 1.46$, $p = .233$, $\eta p2 = .032$, were all non-significant.

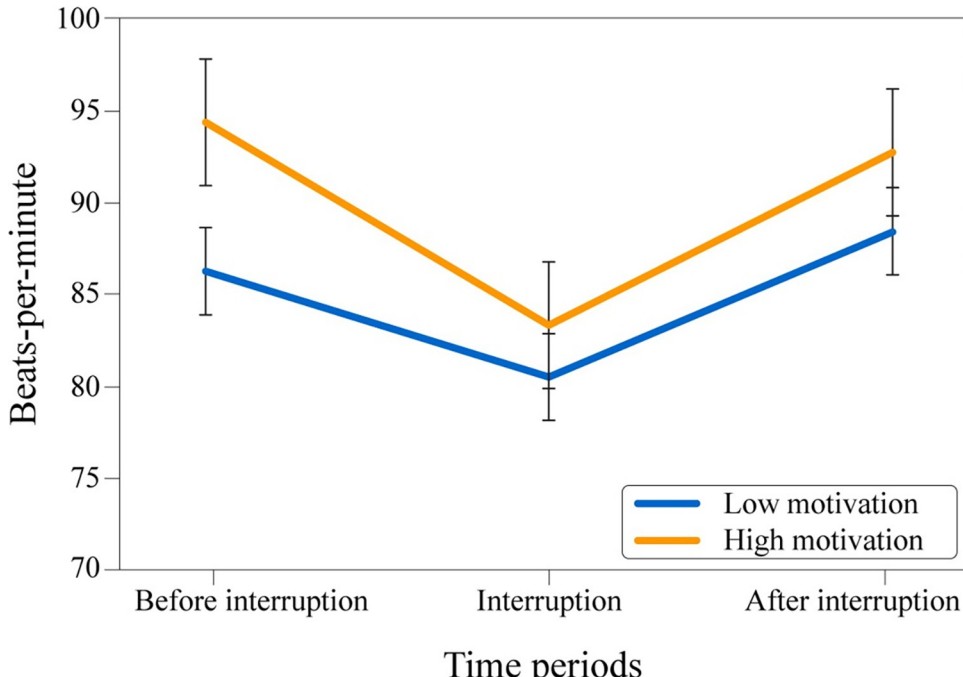

**Fig 8. Mean values of participants' heart rate.** Experimental time periods across conditions.

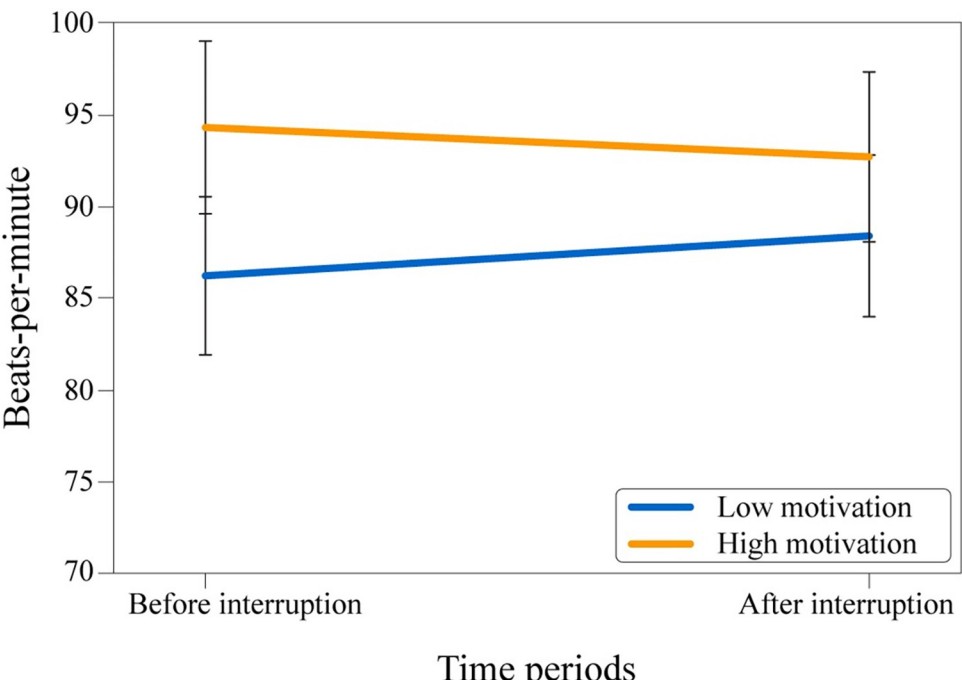

**Fig 9. Mean values of participants' heart rate.** Experimental time periods across conditions.

The results suggested that an observable effect of the heart rate between conditions and interruption was not found. Although the outcomes were not significant, the direction of the effect indicated a decrease in the "after interruption" period for the high motivation group and an increase for the low motivation group, similar to the motivation questionnaire results.

**Density.** Using a 1.5x1.5 square meter measurement area, located half a meter away from the entrance, individual Voronoi density time-series data were calculated and plotted for all participants in each experimental run. The time periods were named "before," "interruption," and "after" for the interruption runs (see Fig 3 for the overall experiment procedure), and "before" and "after" for the without interruption runs. The interruption runs consisted of two high motivation and one low motivation, a total of three runs. Similarly, the without interruption runs consisted of one high motivation and two low motivation, also a total of three runs.

The density plots of the three interruption runs are shown in Fig 10. The density levels vary significantly throughout these three runs, although the pattern is similar. The "before" period shows a steep increase, followed by a plateau. The "interruption" period is characterized by a constant density. During the "after" period, the density decreases quickly or slowly, depending on the level of motivation.

It's worth noting that before the interruption starts, there was always a small time window where participants were allowed to pass through the bottleneck (see Fig 3 for the overall experiment procedure). This time period was colored orange in Fig 10, the same as the "after" time period since it had the same properties. In the high motivation runs, a spike in density occurred before this small time window, presumably due to the excitement of participants at the prospect of reaching their goal after positioning themselves around the bottleneck area. This situation can be seen as a result of high motivation, although the same spike did not occur with the same intensity after the "interruption" time period finished, when participants were free to exit the bottleneck without further interference. A motivational decrease caused by the interruption, as

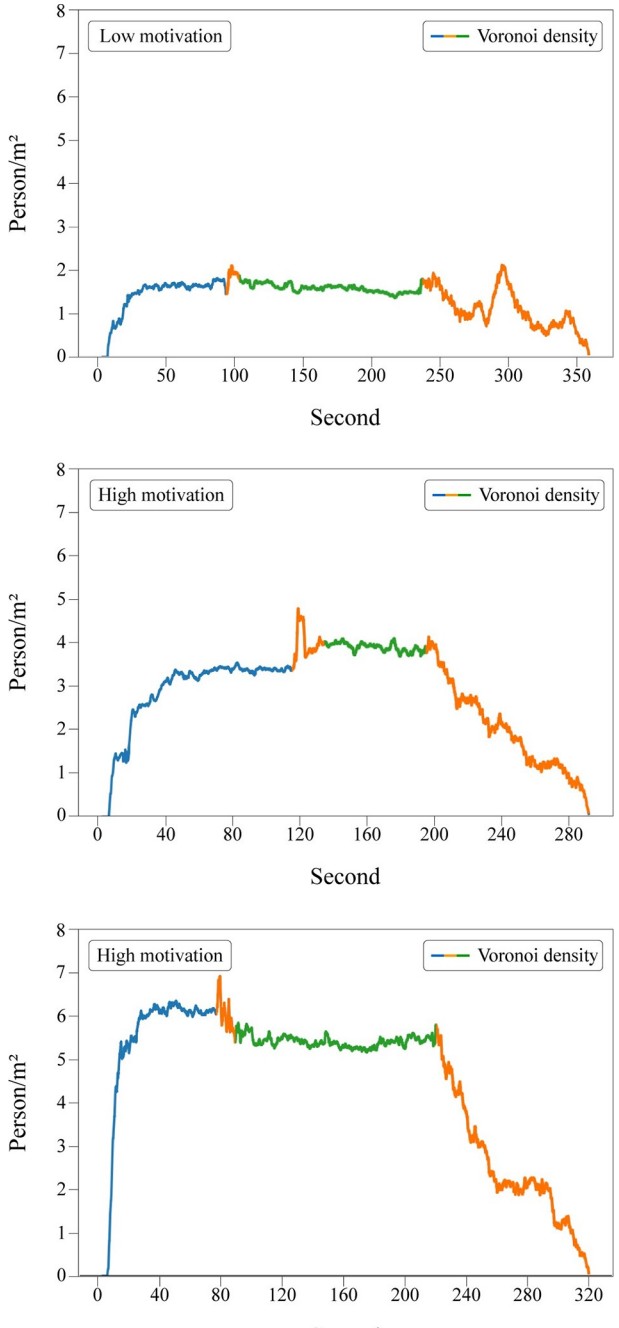

**Fig 10. Merged density plots for interruption runs.** Colors represent the time periods: 'Blue' indicates the time when participants were moving towards the gate while it was still closed. 'Green' indicates the interruption time period. 'Orange' represents the time when the gate was open, and participants were exiting through the bottleneck. Note that the gate was first opened, then closed for the interruption, and then opened again. Both periods were represented as orange accordingly.

found in the questionnaire data (also hinted in the heart rate data), could be the explanation for this situation as well. The same density-increasing environment did not appear after the interruption in high motivation runs. On the other hand, we did not find an effect like this in the low motivation run, neither in the questionnaire data nor in the density plots.

Regarding the without interruption runs, the density plots can be easily interpreted. The density increases throughout the first placement period as participants move to the bottleneck area. A plateau can be observed afterward. The density decreases gradually after participants were instructed to exit the bottleneck (Fig 11).

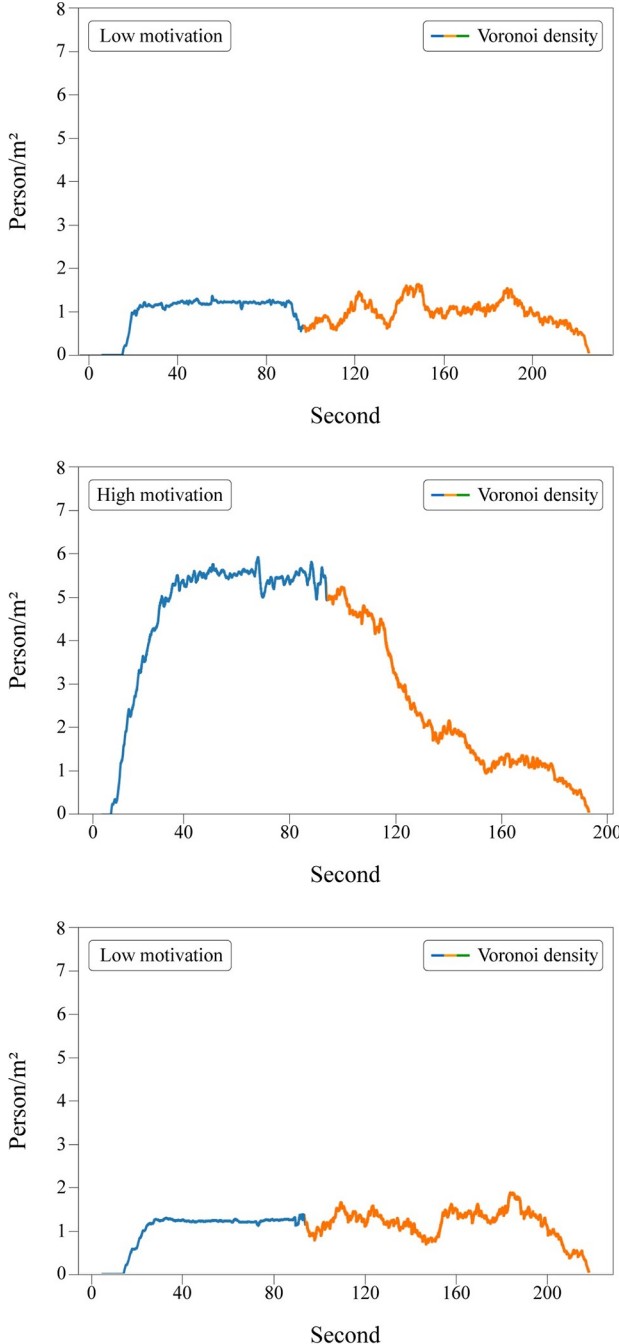

**Fig 11. Merged density plots for without interruption runs.** Colors represent the time periods: 'Blue' indicates the time when participants were moving towards the gate while it was still closed. 'Orange' represents the time when the gate was open, and participants were exiting through the bottleneck.

**Qualitative observations.** It was later decided that explorative-qualitative behavior analysis in video recordings was also to be conducted to capture the complete picture of the experiments and to provide insights for future research. Initial observations regarding the different motivation instructions mainly focused on how people behaved during the interruption. These observations were done during the experiments and written as notes while the experimenters were within a few meters of the crowd, just outside the bottleneck area. It was observed that participants who had received the low motivation instruction were mostly in a relaxed state during the interruption (i.e., yawning, relaxed body postures). On the other hand, participants who had received the high motivation instructions showed more tense body postures and constantly looked around to understand what was going on. One participant was noticeably clicking his pen during the whole interruption period. Regarding the motivation between and after the interruption, it was observed that participants were relatively slower (or relatively lacking interest) when exiting the bottleneck.

Additional observations were made afterward from video recordings. During the low motivation runs (Fig 12), the waiting behavior caused by the interruption did not seem as arousal-inducing. People were not moving regularly; that is, they were mostly standing still apart from occasional head movements to check the environment. They occasionally talked with each other, seemingly trying to make sense of what caused the interruption. The density was not high in the sense that the participants had enough personal space to make themselves comfortable in a crowd situation. It might be the case that the relative relaxation occurred from the

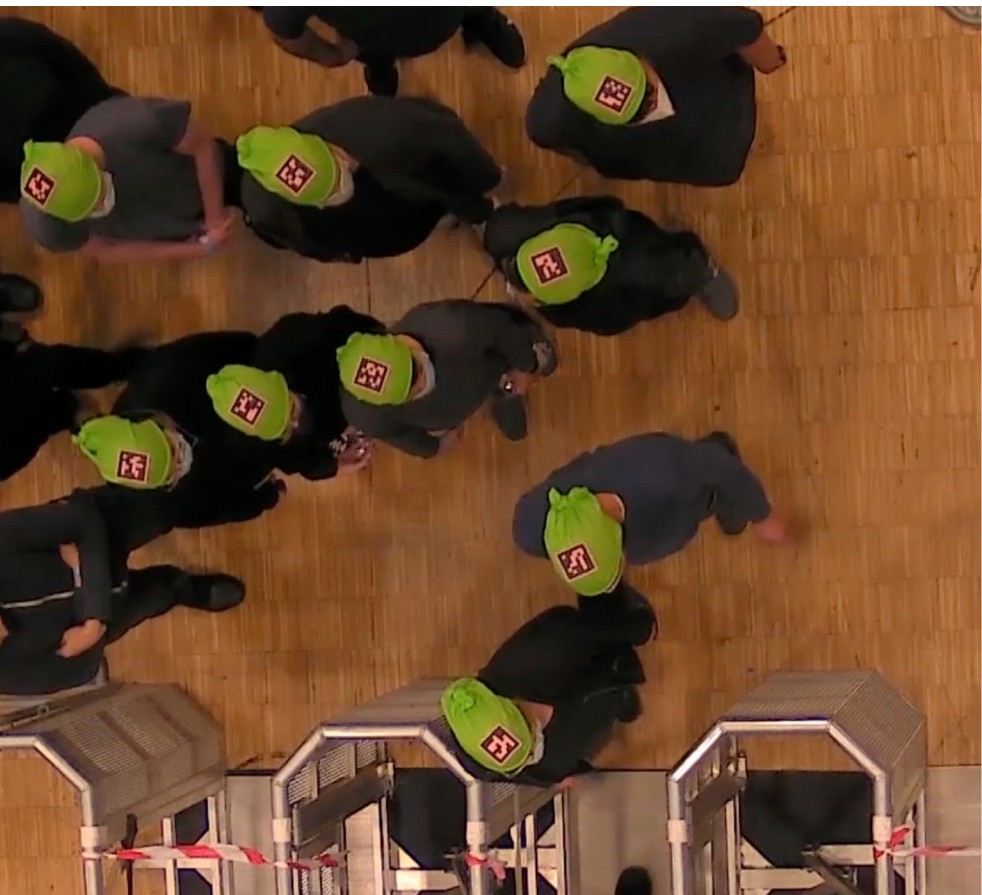

**Fig 12. A screenshot from low motivation experimental runs, during interruption.**

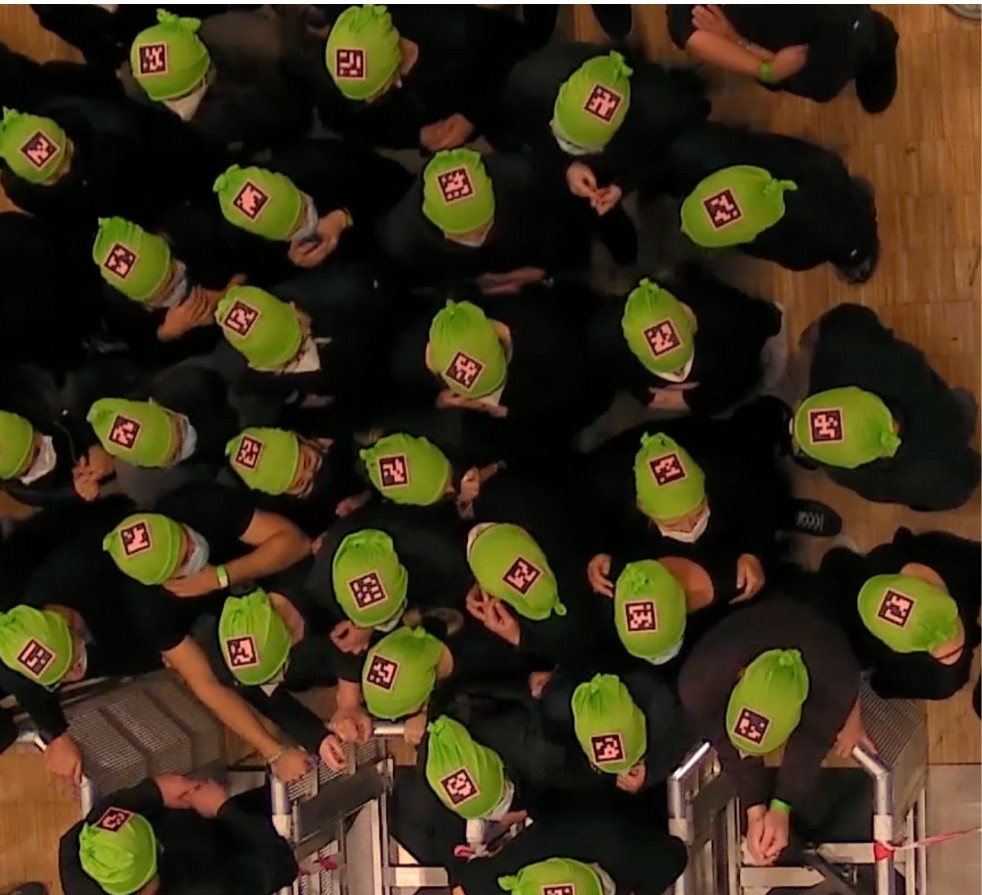

**Fig 13. A screenshot from high motivation experimental runs, during interruption.**

low-density environment, although the low motivation instruction was what created this environment to begin with.

On the other hand, in the high motivation runs (Fig 13), participants moved more frequently, and made more head movements, which can be interpreted as being in a higher state of arousal during the interruption period compared to the low motivation interruption period. The most distinct body posture during the interruption was the "crossed-arms" position. Participants talked to each other more regularly and laughed from time to time, presumably due to the unexpected close-body-contact situation. The density was much higher, and people were close to each other, especially near the bottleneck area. The high-density formation seemingly created a crowded but awkward situation where participants checked the environment more frequently as they tried to make sense of it.

In addition, a decrease in motivation was observed during the high motivation runs. Initially, most participants were relatively fast and pushed others to reach the bottleneck when the experiments started. However, after the interruption, only a small group of participants (approximately 20) who were directly in front of the bottleneck continued to behave in this manner. The rest of the participants created their own personal space and put small distances between themselves and the others during the interruption time period. They waited for these 20 participants to move forward and started to exit the bottleneck one by one without showing excessive acceleration or pushing.

## Discussion

Study 2 explores whether low or high motivation levels cause different emotional or psychophysiological reactions during an interruption event. It was hypothesized that having low motivation when interrupted leads to a 'bored' state but having high motivation when interrupted leads to an 'impatient' state, both psychologically and physically.

The results of the boredom and impatience scales showed that people perceived their respective emotional states in accordance with the initial motivation priming that they were given. If they had high motivation and got interrupted, they associated themselves with an impatience state. Similarly, they associated their respective state with boredom if they had low motivation and got interrupted. Sole impatience and boredom items (I feel bored & I am impatient right now) were also investigated separately. It was found that whether people were bored or impatient mainly depended on their initial motivation instruction, but they were more bored if there was an interruption for the low motivation instruction. The same outcome was also valid for being impatient for high motivation priming, although the effect was smaller. People with high motivation priming were substantially more impatient when there was an interruption and slightly less when there was no interruption. Perhaps this situation was caused by the high motivation priming being more intense than anticipated.

Regarding the heart rate results, no differences were found between the conditions. While this was unexpected, finding the hypothesized outcomes regarding heart rate was difficult to begin with due to the "movement" factor of the experiment. This factor may have had a greater impact on heart rate than the difference between high and low motivation or the interruption. Normally, it would be assumed that an interruption would increase heart rate due to annoyance, but in the context of the experiment, people were forced to move forward during every other time period apart from the interruption period. During the interruption, people were expected to remain still because the path forward was blocked. Taking this into consideration, it was not expected that heart rate would increase, but a non-decrease situation was predicted. However, the results showed that the heart rate meaningfully decreased during the interruption period for all groups. It might be the case that the hypothesis undervalued the effect of the movement.

The hypothesis that the interruption causes a decrease in motivation (as measured before and after the interruption) received expected statistical support. Motivation significantly decreased due to the interruption for the high motivation group, which was consistent with the observations in the experiment area of participants moving slower after the interruption, especially for the high motivation group. Additionally, a sharp increase in density when the bottleneck opened was only measured in high motivation runs before the interruption, not after. However, heart rate data did not show a meaningful decrease for the time period after interruption for the high motivation group–the trend in this direction was not significant.

## General discussion

The present studies investigated different types of interruption events in pedestrian dynamics and game-playing contexts. Firstly, as the literature suggests, it was found that interruption itself affects the respective emotional and motivational state of a person regardless of the type of the interruption, and this particular state often has a negative connotation [15,21,23,24,27]. Following these notions, two different dichotomies, namely early vs. late and low vs. high motivation interruptions, were explored throughout the studies. Early and late interruption yielded no difference, indicating that valuation and goal proximity concepts either had no effect on increasing the annoyance and arousal of the person through an interruption event or the study design failed to create a corresponding environment to produce the effect. However,

low or high motivation showed a meaningful contrast between the emotional and motivational states of people exposed to the interruption. It can be cautiously concluded that the interruption timing has no importance for differentiating the emotional state of a person, but the initial motivation produces varied psychological and psychophysiological outcomes when there is an interruption.

Regarding boredom and impatience assumptions, it was found that these states are indeed distinct, even though state impatience literature does not have detailed theorizing so far [35]. State boredom normally includes impatience as a factor within its discourse [32,33], but it was found that people were keen to perceive themselves as impatient rather than bored in certain situations. If people are highly motivated and get interrupted, they perceive themselves as high aroused or impatient. If people have low motivation and get interrupted, they consequently express their state as being bored or in a state of low arousal.

Continuing with the psychophysiological properties of the hypotheses, the collected data provided mixed results. Few differences were found in the timing of the interruption and the motivational effects of the interruption. Although the direction of the heart rate was as predicted for most cases, the data did not show a statistically significant difference in most situations. People did not have a higher heart rate when they were interrupted in the later stages of their goal pursuit, nor did they have an increase in heart rate when they were interrupted while they were instructed to be highly motivated. However, notably, while the data showed an increase in the heart rate during the interruption period for people in a 'resting' situation (Study 1), the results showed the opposite for people in a 'moving' environment (Study 2). These results show that measuring heart rate with moving participants is challenging.

Lastly, although not studied in Study 1, a decrease in motivation caused by interruption was hypothesized and found in several data from Study 2, suggesting that interruption can impact motivational processes. The participants' motivation decreased for the high motivation group, aligning with what was observed during the experiments. Furthermore, the spike in density after the first bottleneck opening was not repeated after the second opening, suggesting that participants were more active before the interruption compared to after it.

## Limitations

Study 1 had a unique nature due to the Covid-19 situation, which greatly affected experimental research worldwide. Initially, it was decided to hold a crowd experiment, but the experiment context was later changed to individual participation in video game playing because of the restrictions. The early and late interruption conditions should be replicated and put into perspective in a crowd scenario in the future.

Another limitation was encountered for Study 1 while collecting the heart rate data of participants. The collected data were relatively good for the participants in Germany, but there were many instances of data corruption and artifacts from the heart rate data collected from participants in Turkey (see Study 1 - Method—Measures—EcgMove 4). Nearly one-third of the data was unable to be used, presumably because of the heat.

The last limitation of the study is the selected video game: It is possible that the game was not alluring for most participants and therefore did not promote a meaningful annoyance after they were interrupted.

Study 2 did not have any major limitations, almost all experimental runs were held according to the plan, and the collected data did not contain any problems. However, the experimental runs consisted of approximately 100 participants, but we could only provide 20 heart rate devices to the participants due to the limited number of devices at our disposal. We believe that the collected data from 20 devices were enough to represent the rest, but it was still a

limitation that might potentially influence the generalizability of the data. Some data show a clear picture but do not reach the threshold of significance (i.e., Fig 9), and this might partly be due to the comparatively small sample size. Furthermore, the heart rate data showed the effects of movement more strongly than expected, which potentially overlayed other effects such as impatience. One direct solution to avoid this situation would be to create a crowd experiment in which the participants are interrupted while they are already waiting, thereby excluding the movement effect (i.e., an unexpected delay in starting the entrance procedure).

## Future directions and implications

Future studies could expand on various types of interruption. This paper examined whether early or late interruption, or having high or low motivation, has a different impact on the state emotion of the individual who experienced an interruption. A potential future study could explore whether a brief interruption period can elicit a different emotional response compared to a prolonged interruption period. Another possible idea would be to investigate whether interruption and waiting have varying effects on emotional states when individuals are informed about the reasons for the wait.

The results of these studies can also have implications for pedestrian dynamics and traffic contexts. The findings suggest that people's reactions to interruptions can vary, and they may experience states such as boredom and impatience, but it is also known that impatient reactions can pose a risk to traffic safety [35]. People may behave recklessly when it comes to route choices or driving, which can lead to safety issues if interruptions occur during goal pursuit, even if the goal is a routine one such as reaching home. Perhaps the main objective of future policies should be to achieve traffic and pedestrian flow which consist of the minimum number of interruptions possible.

Another valuable aspect to explore in future studies is the sole effect of interruption on motivation, which was statistically observed in Study 2. If this effect of a motivational decrease during interruptions exists in other scenarios, it could be utilized as a method to reduce "motivation" in tense crowd environments. However, the length of the interruption and other factors must be thoroughly studied, as prolonged interruptions could potentially worsen the crowd's tension, as seen in situations such as concerts or sporting events. Furthermore, interrupting only a portion of the crowd could potentially lead to increased density, as individuals in the rear may attempt to push forward. Therefore, it is necessary to make the interruption information accessible to the entire crowd to avoid this unintended consequence.

Broadly viewed, the experiments in this paper contribute to an overall individualistic perspective on motivation. Only individual emotions, bodily reactions, or intentions were measured in both studies. Although a crowd was used in Study 2, it was treated as a large sample size of individuals moving toward a goal. Future studies can potentially investigate the social aspects of interruption events since both individualistic and social effects are intertwined in a crowd context. It is worth exploring what people do to pass the interruption time and how different social contexts affect motivation and emotions during an interruption. As Goffman [45] suggested, awkwardness occurs in social situations where people cannot do anything due to external factors (such as an interruption event). Future research should focus on these interactive aspects of interruptions in crowd dynamics.

## Acknowledgments

We would like to thank Tobias Schrödter (software [Pedpy] assistance), Panar Ege Uesten (image design) and CroMa team (experiment assistance) for their support throughout the study.

## Author Contributions

**Conceptualization:** Ezel Üsten.

**Data curation:** Ezel Üsten.

**Formal analysis:** Ezel Üsten.

**Investigation:** Ezel Üsten, Anna Sieben.

**Methodology:** Ezel Üsten, Anna Sieben.

**Supervision:** Anna Sieben.

**Visualization:** Ezel Üsten.

**Writing – original draft:** Ezel Üsten.

**Writing – review & editing:** Ezel Üsten.

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
