## [Decision Letter · Decision Letter 0]

20 Mar 2023

PONE-D-23-03520Don’t stop me now: Psychological effects of interrupting a moving pedestrian crowd and a video gamePLOS ONE

Dear Dr. Üsten,

Thank you for submitting your manuscript to PLOS ONE. After careful consideration, we feel that it has merit but does not fully meet PLOS ONE’s publication criteria as it currently stands. Therefore, we invite you to submit a revised version of the manuscript that addresses the points raised during the review process. Please submit your revised manuscript by May 04 2023 11:59PM. If you will need more time than this to complete your revisions, please reply to this message or contact the journal office at plosone@plos.org. Please include the following items when submitting your revised manuscript:A rebuttal letter that responds to each point raised by the academic editor and reviewer(s). You should upload this letter as a separate file labeled 'Response to Reviewers'.A marked-up copy of your manuscript that highlights changes made to the original version. You should upload this as a separate file labeled 'Revised Manuscript with Track Changes'.An unmarked version of your revised paper without tracked changes. You should upload this as a separate file labeled 'Manuscript'.

We look forward to receiving your revised manuscript.

Kind regards,

Charitha Dias

Academic Editor

PLOS ONE

Journal Requirements:

Additional Editor Comments :

Two experts in the field of crowd dynamics have reviewed your manuscript and provided comments. There are some improvements that both reviewers would like to see in the figures. A reviewer would like to see some additional discussions regarding data extraction from the raw data. In addition, a discussion of the choice of the sample should also be included.

Reviewers' comments:

Reviewer's Responses to Questions

**Comments to the Author**

1. Is the manuscript technically sound, and do the data support the conclusions?

Reviewer #1: Yes

Reviewer #2: Yes

2. Has the statistical analysis been performed appropriately and rigorously? 

Reviewer #1: Yes

Reviewer #2: Yes

3. Have the authors made all data underlying the findings in their manuscript fully available?

Reviewer #1: Yes

Reviewer #2: Yes

4. Is the manuscript presented in an intelligible fashion and written in standard English?

Reviewer #1: No

Reviewer #2: Yes

5. Review Comments to the Author

Reviewer #1: This manuscript focuses on studying the effects of interruptions on individuals from a psychological and crowd dynamics perspective, which is interesting and practical. Experiments and numerical analysis of the results are presented in a very clear way. I would like to make some observation about the work.

1)Abstract should be improved to provide a better overview of the study.

2)Whether the heterogeneity of the research sample (Germany and Turkey) will affect the research results, have you compared the differences of results between the two different countries in study 1?

3)The reviewer advised that the diagram should be put in the text rather than appendix, and the alignment of manuscript is unsightly, please modify the whole manuscript.

4)In line 212, the author mentioned that “The raw data of each participant were then cut manually into their respective time periods”. Why did use to manual way? What is the criterion for cutting?

5)In line 418, the author mentioned that “20 participants from each group (a total of 60) were selected and instructed to wear the device before the experiment began”. Why did only select 20 participants? What is the criterion for selecting 20 participants?

6)The Fig. 3 not present clear information, the reviewer advised to draw the picture for convenience of expressing content.

7)The lanuage of the manuscript should be polished carefully. There are a lot of mistakes inside

Reviewer #2: This paper discusses the psychophysiological aspects of interruptions in different contexts, both in individual and collective situations. Design of the experiments was largely disrupted due to the COVID-19 pandemics and so it is divided into two studies. In the first experiment participants are interrupted while playing a video game with the aim to study whether an interruption shortly before reaching an important goal (within the scope of the game) would be different than an interruption occurring shortly after starting to play. In the crowd experiments the authors studied the effect of motivation by checking whether an interruption in a highly motivated crowd would be different than in a low motivated crowd. To check their hypothesis both questionnaire and medical devices were used and, for crowd experiments, information from videos were also analyzed.

Results generally show that interruptions do affect the psychological state of people, but in a way a bit different than what originally expected. Goal pursuit apparently play a minimal role, but motivation was found to be relatively important to determine the changes observed during interruption. Also, the study showed opportunities and limitations in the use of medical devices (hearth rate here) to study crowd conditions. In that regard hearth rate was difficult to evaluate based on technical limitation (high temperatures in Turkey) and motion during crowd experiments.

The study covers an original topic which has been investigated only little and would deserve more research. The paper is well written, follows a logical structure and all details needed to replicate the results are provided. Statistical testing is performed following solid methods and reported in accordance to guideline relative to psychological studies.

As such, given the reasons provided above, I believe the manuscript can be accepted with minimal modifications whose details are given below.

1. Line 128: Is “distortive” what you want to say. Makes sense, but maybe “disrupting” was what you meant. Not sure; please check.

2. Line 161: “being nearly at the would”  “being nearly at the end”

3. Line 209: I might be wrong but the fact that participants had to play a game is discussed here taking for granted that readers would know that. But I believe the fact that participants were playing a game is only introduced later (except for the abstract).

4. Line 250: I was wondering; why didn’t you ask to fill the questionnaire during the interruption? I am not criticizing the procedure, it is perfect. Personally, I would have asked them to start filling the questionnaire while you “fix the technical” issue, just to have an idea on how they felt. Then you can ask them again to fill the questionnaire at the end. Of course the questionnaire during the interruption need to be a bit different than the final one, but with some deception you may gain a better insight on their state of mind during the interruption. Maybe it could be an idea for a follow-up experiment; again, not criticizing the work.

5. Line 262: “around ten participants”; maybe you can provide the percentage, just to have a better idea although the total number is given.

6. Lines 373-379: Maybe you can add something like “details will follow” just to let readers know that this aspect will be discussed in detail later.

7. Lines 393-394: I guess you had homosexual, transgender or people not providing their gender (unidentified). Maybe you can add the percentage for “other” just to show that it comes to 100% and there is no mistake in reporting.

8. Lines 396-397: “without” is repeated.

9. Line 414: I think you did have a deception because people did not know that the interruption was the goal of the experiment itself. So, I am not sure what you mean here by saying “no deceptive items were used.”

10. Fig. 3 looks black and white to me although it is stated that orange is used.

11. Fig. 10 was quite unclear to me at first. I would use a different color for the orange in the middle. Using the same color is a bit confusing. Also, maybe you can add the condition (motivation) as a text in the images instead of having it written in the caption. Having some text in the white space of the figure would help reading them. Also, I guess having seconds instead of frames is better (although you wrote that 1 s = 50 frames).

12. Fig. 11, similar remarks on labeling and x-axis.

13. Lines 817-819: I would be a bit careful in suggesting to stop a crowd considering accidents which occurred in the past. If a crowd has to be stopped the whole crowd must be informed. Having only a small part of the crowd stop moving with the rest in movement can be dangerous in some situations.

14. Graphs: I would better change them a bit just to avoid them being in the standard Excel format. I am not at all against using Excel which is a valid software for visualizing results, but having a choice of color which is not the standard one could help presenting the study in a more “professional” way. Just my personal advice anyway, there is no problem at all in the way it is presented now.

15. Final comment: I loved the title! I was singing while doing the review.

6. PLOS authors have the option to publish the peer review history of their article (what does this mean?). If published, this will include your full peer review and any attached files.

Reviewer #1: No

Reviewer #2: **Yes: **Claudio Feliciani

---

## [Author Response · Author response to Decision Letter 0]

2 May 2023

Editor: Regarding the "additional requirements," all three points have been addressed in the "Response to Reviewers" file. Two additional notes (1- proofreading by an AI, and 2 - a correction in the funding number for Study 2) have also been added.

Reviewer 1: I have edited all the points addressed within the manuscript. Answers have been provided in the "Response to Reviewers" file. A general proofreading has been completed.

Reviewer 2: I have edited all the points addressed within the manuscript. Answers have been provided in the "Response to Reviewers" file. Figures have been revised and recreated.

---

## [Decision Letter · Decision Letter 1]

8 Jun 2023

Don’t stop me now: Psychological effects of interrupting a moving pedestrian crowd and a video game

PONE-D-23-03520R1

Dear Dr. Üsten,

We’re pleased to inform you that your manuscript has been judged scientifically suitable for publication and will be formally accepted for publication once it meets all outstanding technical requirements.

Kind regards,

Charitha Dias

Academic Editor

PLOS ONE

Additional Editor Comments (optional):

Reviewers' comments:

Reviewer's Responses to Questions

**Comments to the Author**

1. If the authors have adequately addressed your comments raised in a previous round of review and you feel that this manuscript is now acceptable for publication, you may indicate that here to bypass the “Comments to the Author” section, enter your conflict of interest statement in the “Confidential to Editor” section, and submit your "Accept" recommendation.

Reviewer #2: All comments have been addressed

2. Is the manuscript technically sound, and do the data support the conclusions?

Reviewer #2: Yes

3. Has the statistical analysis been performed appropriately and rigorously? 

Reviewer #2: Yes

4. Have the authors made all data underlying the findings in their manuscript fully available?

Reviewer #2: Yes

5. Is the manuscript presented in an intelligible fashion and written in standard English?

Reviewer #2: Yes

6. Review Comments to the Author

Reviewer #2: The author(s) kindly answered to all my concerns and the manuscript has remarkably improved (especially in regard to the visual aspects); scientific content and language was already good in my opinion. I believe the manuscript can be accepted for publication in the present form. Congratulations to the authors.

7. PLOS authors have the option to publish the peer review history of their article (what does this mean?). If published, this will include your full peer review and any attached files.

Reviewer #2: **Yes: **Claudio Feliciani

---

## [Editor Report · Acceptance letter]

7 Jul 2023

PONE-D-23-03520R1 

Don’t stop me now: Psychological effects of interrupting a moving pedestrian crowd and a video game 

Dear Dr. Üsten:

I'm pleased to inform you that your manuscript has been deemed suitable for publication in PLOS ONE. Congratulations! Your manuscript is now with our production department. 

Kind regards, 

on behalf of

Dr. Charitha Dias 

Academic Editor

PLOS ONE